# Artificial intelligence in pancreatic intraductal papillary mucinous neoplasm imaging: A systematic review

**Muhammad Ibtsaam Qadir**[1], **Jackson A. Baril**[2], **Michele T. Yip-Schneider**[2],
**Duane Schonlau**[3], **Thi Thanh Thoa Tran**[2], **C. Max Schmidt**[2,4], **Fiona R. Kolbinger**[1,5*]

**1** Weldon School of Biomedical Engineering, Purdue University, West Lafayette, Indiana, United States of America, **2** Division of Surgical Oncology, Department of Surgery, Indiana University School of Medicine, Indianapolis, Indiana, United States of America, **3** Department of Radiology, Indiana University School of Medicine, Indianapolis, Indiana, United States of America, **4** Department of Biochemistry and Molecular Biology, Indiana University School of Medicine, Indianapolis, Indiana, United States of America, **5** Department of Visceral, Thoracic and Vascular Surgery, University Hospital and Faculty of Medicine Carl Gustav Carus, TUD Dresden University of Technology, Dresden, Germany

\* fkolbing@purdue.edu

## Abstract

Based on the Fukuoka and Kyoto international consensus guidelines, the current clinical management of intraductal papillary mucinous neoplasm (IPMN) largely depends on imaging features. While these criteria are highly sensitive in detecting high-risk IPMN, they lack specificity, resulting in surgical overtreatment. Artificial Intelligence (AI)-based medical image analysis has the potential to augment the clinical management of IPMNs by improving diagnostic accuracy. Based on a systematic review of the academic literature on AI in IPMN imaging, 1041 publications were identified of which 25 published studies were included in the analysis. The studies were stratified based on prediction target, underlying data type and imaging modality, patient cohort size, and stage of clinical translation and were subsequently analyzed to identify trends and gaps in the field. Research on AI in IPMN imaging has been increasing in recent years. The majority of studies utilized CT imaging to train computational models. Most studies presented computational models developed on single-center datasets (n = 11,44%) and included less than 250 patients (n = 18,72%). Methodologically, convolutional neural network (CNN)-based algorithms were most commonly used. Thematically, most studies reported models augmenting differential diagnosis (n = 9,36%) or risk stratification (n = 10,40%) rather than IPMN detection (n = 5,20%) or IPMN segmentation (n = 2,8%). This systematic review provides a comprehensive overview of the research landscape of AI in IPMN imaging. Computational models have potential to enhance the accurate and precise stratification of patients with IPMN. Multicenter collaboration and datasets comprising various modalities are necessary to fully utilize this potential, alongside concerted efforts towards clinical translation.

**Data availability statement:** All relevant data are within the manuscript and its Supporting Information files.

**Funding:** CMS and FRK receive support from the Evan and Sue Ann Werling Pancreatic Cancer Research Fund and the Indiana Clinical and Translational Sciences Institute funded, in part, by Grant Number UM1TR004402 from the National Institutes of Health, National Center for Advancing Translational Sciences, Clinical and Translational Sciences Award. Furthermore, FRK receives support from the German Cancer Research Center (CoBot 2.0) and the Joachim Herz Foundation (Add-On Fellowship for Interdisciplinary Life Science). The content is solely the responsibility of the authors and does not necessarily represent the official views of the National Institutes of Health.

**Competing interests:** I have read the journal's policy and the authors of this manuscript have the following competing interests: FRK declares unpaid advisory roles for Perspectum, Inc., Oxford, UK; Radical Healthcare, Inc., San Francisco, CA; and the Surgical Data Science Collective (SDSC), Washington, DC. CMS declares an unpaid advisory role for Perspectum, Inc., Oxford, UK. All other authors declare no competing interests.

## Author summary

Pancreatic cysts, such as intraductal papillary mucinous neoplasms, are common and can sometimes develop into pancreatic cancer. Currently, clinicians plan the treatment of these cysts mainly through imaging, using established guidelines. While these guidelines show adequate performance in identifying high-risk patients, they often struggle to accurately determine which cysts are harmless. Artificial intelligence shows promise in improving the diagnosis and aiding clinicians in treatment planning for these cysts. Our review of recent scientific research shows an increasing number of studies exploring the potential of artificial intelligence in this area. Most of these studies use computational models that are trained on small imaging datasets from one hospital, and they primarily focus on classifying different types of cysts. The research suggests that artificial intelligence could help clinicians make better decisions about patient care. However, to fully realize this potential, researchers need to develop models using larger, more diverse datasets from multiple hospitals.

## Introduction

Pancreatic cancer is one of the most aggressive cancers, with a 5-year overall survival rate that remains under 15% [1,2]. Pancreatic ductal adenocarcinoma (PDAC) is the most prevalent solid tumor of the pancreas and constitutes more than 85% of all pancreatic cancer incidences [3,4]. Over one-third of patients receive a late diagnosis at an incurable stage [5], and only approximately 20% present with surgically resectable disease at the time of diagnosis [6]. Some pancreatic cysts, often incidentally detected on imaging [7], are potential precursors to pancreatic cancer, presenting a window of opportunity for early detection and cancer prevention. The most common type of pancreatic cyst is intraductal papillary mucinous neoplasm (IPMN) [8]. About 10% of the general population over 50 years of age is estimated to carry IPMN [9]. The frequency of invasive carcinoma in resected branch duct (BD-)IPMN and main duct (MD-)IPMN ranges from 10 to 45% [10–12] and from 40 to 75% [12–14], respectively. In patients with BD-IPMN, the 5-year and 15-year malignant progression rates are 3.3% and 12%, respectively [15]. Of all PDACs, 8–10% are estimated to derive from IPMN [16,17].

The management of IPMN primarily depends on imaging features, cytology, and clinical variables such as CA19–9 serum levels and pancreatitis [9,18]. Patients with MD-IPMN and pancreatic duct dilation to a diameter of 10 mm or more or a nodule of at least 5 mm are generally recommended surgical resection; in patients with BD-IPMN, imaging features such as main duct dilation to a diameter of 5–9 mm, enhancing mural nodule less than 5 mm, cyst size ≥ 30 mm or a cyst growth rate ≥ 2.5 mm/year are considered indicative of higher malignant potential warranting consideration for surgical resection [12]. Patients carrying BD-IPMN without such "worrisome features" are recommended surveillance every 6 or 12 months, including

imaging and a clinical examination [12]. Pancreatic surgery is associated with substantial risks, including mortality [19] and morbidity [20] such as post-pancreatectomy diabetes mellitus [21]. Therefore, recommendations for surgical resection are generally determined by experts who evaluate each case using all available diagnostic modalities [22]. The Fukuoka [12] and Kyoto international consensus guidelines [23] represent the gold standard for IPMN management; however, while they are sensitive, they lack specificity [24], Consequently, patient risk stratification is often not accurate, either overestimating malignant transformation risk resulting in surgical overtreatment [25,26], or underestimating the malignant potential, resulting in surveillance of malignant IPMN (undertreatment) [27].

Artificial Intelligence (AI), particularly deep learning, can detect characteristic patterns in large medical imaging datasets, such as radiological, histopathological, and endoscopic imaging, that are beyond what humans can identify [28,29]. Similarly, traditional natural language processing (NLP) models and, more recently, large language models (LLMs), facilitate the interpretation of written reports, which may support clinical decision-making and personalization of patient care [30,31]. The complex management of IPMN patients is an ideal use case for AI-based computational tools, which could contribute to clinical decision-making and risk stratification based on available data, particularly imaging as the primary non-invasive source of information about the malignant potential of IPMNs.

This systematic review provides an overview of the scientific literature on AI in IPMN imaging. We summarize relevant clinical and technological aspects of published studies, including patient selection criteria and cohort sizes, prediction targets, data modalities, methodological characteristics, performance metrics, translational considerations, and data availability. By highlighting the research trends and gaps, this review aims to showcase the potential of computational methods in the context of the clinical management of IPMN.

## Methods

### Search strategy

We performed this systematic review following the PRISMA (Preferred Reporting Items for Systematic Reviews and Meta-Analyses) [32] and AMSTAR (A MeaSurement Tool to Assess systematic Reviews) [33] guidelines. To cover the interdisciplinary literature on AI in IPMN imaging, our search strategy included four literature databases (PubMed, IEEE Xplore, Embase, and CINAHL [Cumulative Index to Nursing and Allied Health Literature]), and three preprint servers (ArXiv, medRxiv, and bioRxiv). Three authors with experience in medical AI research (MIQ, JAB, FRK) and a librarian focusing on biomedical literature contributed to the development of the search strategy. Medical subject headings (MeSH) terms were used to specify the search term.

PubMed, IEEE Xplore, Embase, CINAHL, and ArXiv were searched on January 31, 2024, without language restrictions, for literature published since database inception, using the following search term: (("Pancreatic Neoplasms"[MeSH Terms]) OR ("Pancreatic Cyst"[MeSH Terms]) OR ("Pancrea* Neoplasm*"[Title/Abstract])) AND (("Diagnostic Imaging"[MeSH Terms]) OR (diagnosis[MeSH Terms]) OR (Record*[Title/Abstract]) OR (Tomography[Title/Abstract]) OR (Radiolog*[Title/Abstract]) OR (methods[MeSH Terms]) OR ("Electronic Medical Record"[Title/Abstract]) OR ("EMR"[Title/Abstract]) OR (Report*[Title/Abstract])) AND (("Artificial Intelligence"[MeSH Terms]) OR ("Deep Learning"[Title/Abstract]) OR ("Natural Language Processing"[MeSH Terms]) OR ("neural network*"[Title/Abstract]) OR ("Machine Learning"[Title/Abstract]) OR (AI[Title/Abstract])). The format of the search term was modified per the databases' requirements. MedRxiv and bioRxiv were searched on January 31, 2024, using the following, shorter search term to comply with the character restrictions of the respective search tools: ("Learning" OR "Artificial Intelligence") AND ("Pancreas" OR "Pancreatic") AND ("Imaging" OR "EMR").

### Study selection

Duplicate studies were removed automatically and manually. All studies were independently screened by two raters: one physician (JAB) with three years of experience in IPMN management and research and one engineer (MIQ) with two

years of experience in AI and computational medical image analysis, using the Covidence platform (Covidence systematic review software, Veritas Health Innovation, Melbourne, Australia, https://www.covidence.org/). Screening results were blinded until the completion of each reviewer's individual screening. The screening process was carried out in two iterations: First, screening based on title and abstract, and second, full-text screening. In each iteration, conflicts between the two reviewers were resolved by independent assessment of a third reviewer (FRK).

The inclusion criteria were: (i) analysis of IPMN (and synonyms: mucinous cyst, mucinous cystic lesion) imaging, including the analysis of broader patient cohorts with pancreatic cystic lesions if the reported findings were relevant to IPMN or if data from patients with IPMN was included in the work, (ii) development of computational models (machine learning, deep learning, or NLP) for the analysis of medical images and/or imaging reports, (iii) imaging modality being Endoscopic Ultrasonography (EUS) or tomography, i.e., Magnetic Resonance Imaging (MRI) or Computed Tomography (CT), (iv) original research, (v) analysis of human data, and (vi) the article being in English. Non-original articles and publications (i) not analyzing any data from patients with IPMN or (ii) not developing computational models for data analysis (i.e., statistical methods, manually curated imaging features) were excluded.

### Data extraction and analysis

One reviewer with expertise in AI and deep learning-based medical image analysis (MIQ) extracted the following data from each included study based on a data extraction protocol (S1 Table): (i) year of publication, (ii) prediction targets, (iii) ground truth source, (iv) number of centers in the study, (v) patient selection criteria, (vi) total number of patients on which the AI model was developed, (vii) number of patients with IPMN on which the AI model was developed, (viii) number of patients in the training set, (ix) number of patients in test/validation set, (x) validation procedure, (xi) data/imaging type, (xii) computational methodology, (xiii) model performance, (xiv) stage of clinical translation, and (xv) data availability. To assess the risk of bias, two reviewers (MIQ and JAB) independently evaluated the studies using PROBAST [34] (prediction model risk of bias assessment tool), with a third reviewer (FRK) to resolve discordances. The overall risk of bias for each study was categorized as low, high, or unclear, based on the evaluations across four domains: participants, predictors, outcomes, and analysis.

We systematically categorized the studies into four use cases based on the prediction targets: (i) detection, (ii) segmentation, (iii) differential diagnosis, and (iv) risk stratification. Detection encompasses the identification of the presence of pancreatic cysts, including IPMN. Segmentation refers to the precise localization of the pancreatic cyst. This use case included publications developing a segmentation method for pixel- or voxel-wise delineation of IPMN or other pancreatic cysts in the radiological images. Differential diagnosis concentrates on differentiating between types of pancreatic cysts, including cysts other than IPMN. Risk stratification includes classifying the IPMN and other pancreatic cysts into IPMN with low-grade dysplasia (benign cysts) or IPMN with high-grade dysplasia (malignant cysts). To provide a comprehensive account of the previous research, we used descriptive statistics to examine the data for each category. Key findings are summarized to highlight trends and gaps in the current literature.

### Study registration

This systematic review was prospectively registered at OSF on January 4, 2024 (https://doi.org/10.17605/OSF.IO/NVGXB) and at PROSPERO on August 13, 2024 (CRD42024575960). The protocol was not amended or changed.

## Results

### Search results

The search strategy, when queried across all the databases and preprint servers, resulted in 1041 publications (PubMed: n = 558, Embase: n = 199, CINAHL: n = 51, IEEE Xplore: n = 45, bioRxiv: n = 137, medRxiv: n = 47, arXiv: n = 4, Fig 1). After the removal of duplicates (n = 125), 916 unique publications were screened.

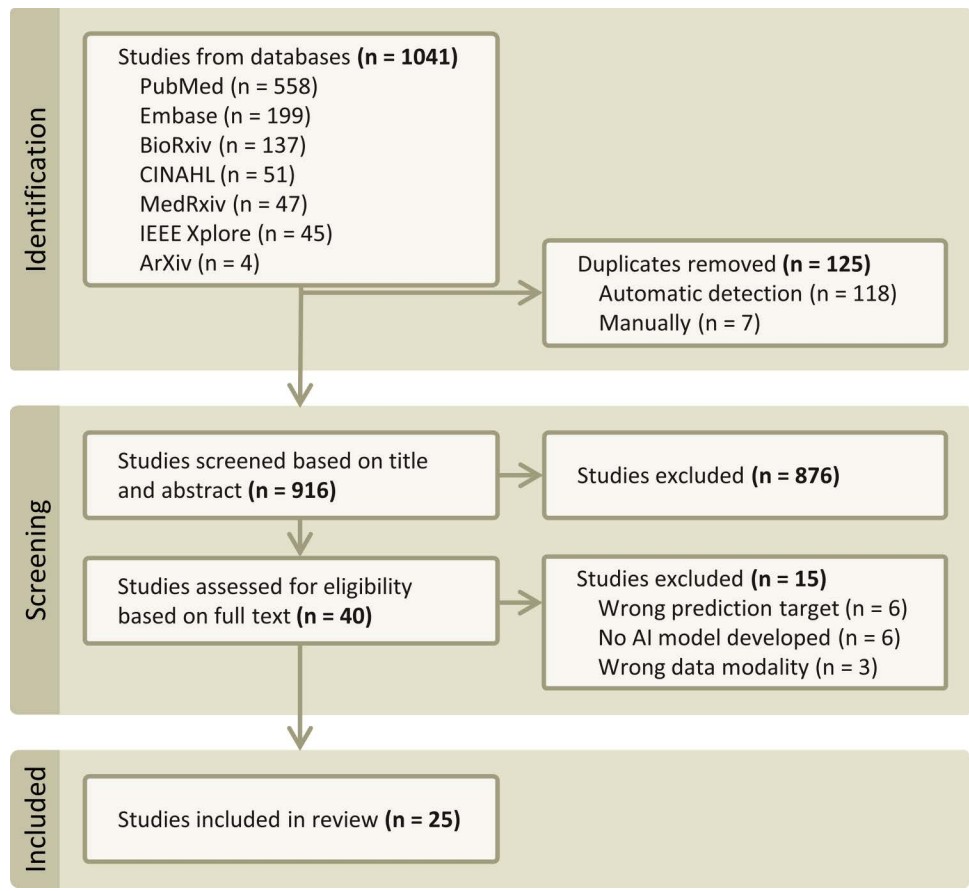

**Fig 1. Flowchart of the systematic review and meta-analysis according to the PRISMA 2020 statement for reporting systematic reviews.** Based on the systematic review of the publications on AI in IPMN imaging, 25 articles published between 2018 and 2023 were included in this analysis.

Based on title and abstract screening, 40 studies were selected for full-text screening. A total of 15 publications were excluded based on their prediction target (n = 6), data modality (n = 6), or for not developing any AI model (n = 3). Specifically, studies involving deep learning-based MRI reconstruction were excluded, as these fell outside the scope of diagnosis or risk stratification of IPMN. Additionally, studies using imaging modalities other than MRI, CT, or EUS were excluded to align with consensus guidelines for IPMN management [12,23] and the American College of Radiology's (ACR) Appropriateness Criteria for pancreatic cysts [35]. This resulted in the inclusion of 25 publications in the systematic review (Fig 1).

### Study characteristics

All included studies were retrospective. Studies were categorized based on the analyzed imaging modality. The majority of publications on AI in IPMN imaging (n = 15, 60%) analyzed CT imaging, one-third of the studies (n = 8, 32%) employed MRI imaging, while only a few (n = 2, 8%) were based on EUS imaging (Fig 2).

We scrutinized the publication trend over time, both collectively and for each prediction target. All included studies were conducted between 2018 and 2023. The number of publications increased over time, with more than two-thirds (n = 17, 68%) of all the studies published within the last three years (2021–2023), and the highest number of studies published in 2023 (n = 8, 32%, Fig 2).

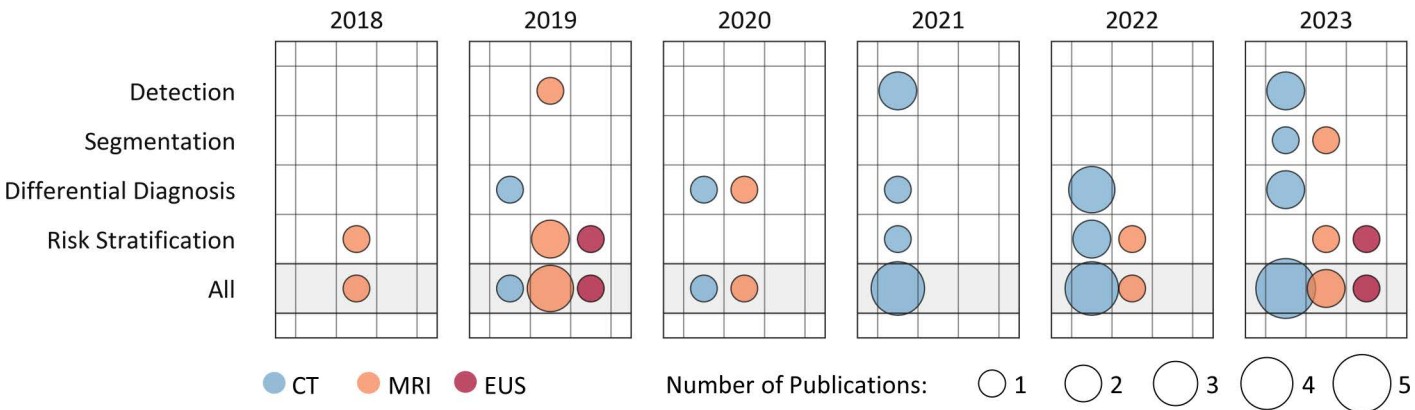

**Fig 2. Published studies by year of publication and prediction target.** The studies are further categorized by the imaging modality used. The radius of each circle is proportional to the number of publications, with colors indicating imaging modality.

Based on the prediction target, approximately one-fourth of the included studies (n = 5, 20%) focused on detection, a couple of studies (n = 2, 8%) on segmentation, while the majority of the studies performed differential diagnosis (n = 9, 36%) and risk stratification (n = 10, 40%) of pancreatic cysts and IPMN (Fig 3).

## Current landscape and clinical translation of AI in IPMN imaging

Regarding the stage of clinical translation, over two-thirds of the studies (n = 17, 68%) conducted internal validation, 7 (28%) carried out external validation, and only one study (4%) proceeded to a prospective clinical evaluation (Fig 3A). Nearly half of the studies (n = 11, 44%) were single-center studies, and one-third (n = 8, 32%) were multicenter studies, with most of these being two-center studies (n = 5, 20%). The remaining studies (n = 6, 24%) did not document the number of centers explicitly (Fig 3B, Table 1).

The majority of the studies (n = 18, 72%) investigated cohorts of less than 250 patients. Three studies (12%) included over 1000 patients in total. Most of the studies had an IPMN patient population of less than 100 (n = 13, 52%), and ten studies (40%) were within the range of 100–200, while only one study (4%) included over 200 patients with IPMN (Fig 3B). Only one-third of the included studies (n = 8, 32%) mainly focused on IPMN, with the remaining studies (n = 17, 58%) including IPMN patients as a subset of total patients (S2 Table). In the studies with a large overall patient cohort size, i.e., over 1000, only a small subset of patients, under 7%, had a pathologically confirmed IPMN (Fig 3B, Table 1). The diagnosis of IPMN or other pancreatic cysts was established through surgical pathology reports in most of the studies (n = 21, 84%), while in the other studies (n = 4, 16%) annotations provided by radiologists were considered as ground truth labels (Table 1).

The methodologies exhibit heterogeneity, yet a majority of studies used Convolutional Neural Networks (CNN) to develop the prediction model. About half of the included studies evaluate these models on independent datasets, and an almost equal number of studies present cross-validation results (Table 1).

Studies focused on IPMN/cyst detection predominantly employed the U-Net architecture for the pancreas region of interest (ROI) segmentation. Alternatively, some studies used a sequential model setup with CNN-based feature extraction followed by machine learning classifiers for final detection. Reported sensitivity across these studies generally exceeded 80%. Variants of the U-Net architecture optimized specifically for pancreatic cyst segmentation were commonly utilized for segmentation tasks. The reported dice similarity coefficient (DSC) for segmentation ranged from 0.42 to 0.80. For differential diagnosis, approximately half of the studies (n = 4) extracted radiomic features and utilized machine learning

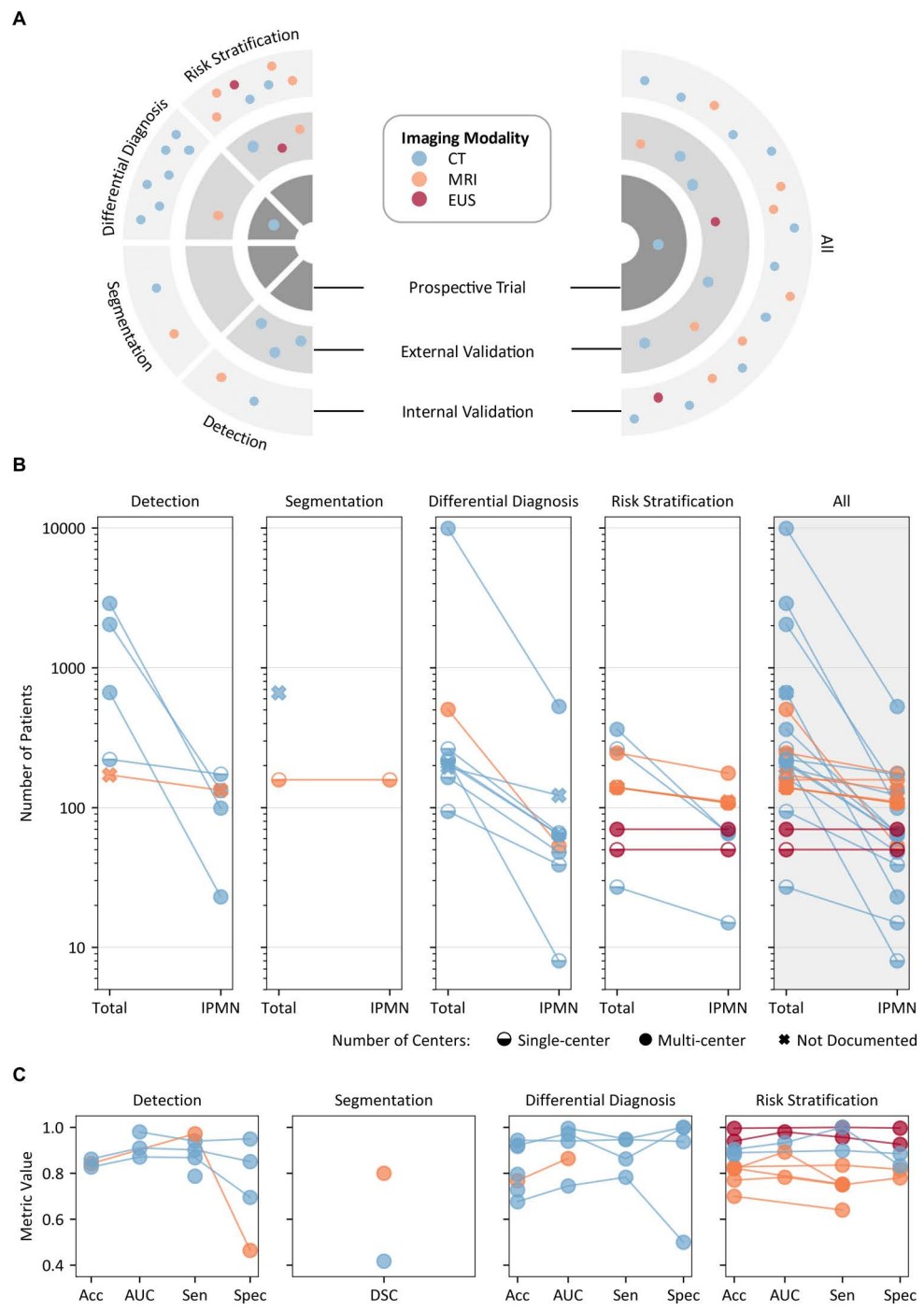

**Fig 3. Overview of research on AI in IPMN imaging. (A)** Distribution of publications on AI in IPMN imaging based on prediction target (IPMN/cyst detection, cyst segmentation, differential diagnosis, risk stratification) and stage of clinical translation: internal validation (monocentric studies), external validation (multicentric studies), prospective clinical evaluation. Each dot represents a single publication, with the dot color indicating the imaging modality used. **(B)** Size of the total patient cohort and the patient cohort with IPMN. Colors indicate the imaging modality used, and symbol shapes indicate the number of centers involved in the study. One study evaluating segmentation did not document the number of IPMN patients in the underlying cohort. **(C)** Performance metrics - Accuracy (Acc), Area Under the Receiver Operating Curve (AUC), Sensitivity (Sen), and Specificity (Spec) - for the proposed model across prediction targets in each study. Colors represent the imaging modalities used. For segmentation, the DSC is presented. Not all studies reported every metric.

**Table 1. Summary and study characteristics of publications included in this systematic review.**

| Study ID | Prediction Targets | Ground Truth | Number of Centers | Number of Patients AI was Developed on — Total | Number of Patients AI was Developed on — IPMN | Number of Patients — Train | Number of Patients — Test | Validation Procedure | Imaging Modality | Methodology | Performance Metrics | Stage of Clinical Translation | Overall ROB |
|---|---|---|---|---|---|---|---|---|---|---|---|---|---|
| Abel 2021 [36] | Detection | Rad | 1 | 221 | 173 | 221 | ND | CV | CT | Two-step nnUnet architecture | Mean sensitivity: 78.8±0.1% | Internal validation | High |
| AbiNader 2023 [37] | Detection | Path | 9 | 2890 | 99 | 2134 | 756 | CV & TS | CT | 3D nnUNet and logistic regression | AUC: 0.98 (overall); 0.98 (IPMN) | External validation | Unclear |
| Cao 2023 [38] | Differential Diagnosis | Path | 10 | 9939 | 529 | 3208 | 6731 | CV & TS | CT | Multitask CNN, dual-path memory transformer | AUC: 0.996 (lesion detection); 0.987 (PDAC vs non-PDAC; Acc: 79.6% (differential diagnosis) | Prospective clinical trial | Low |
| Chu 2022 [39] | Differential Diagnosis | Path | 1 | 214 | 64 | 214 | ND | CV | CT | RM using RF for cyst classification | AUC: 0.942 (IPMN); 0.940 (overall) | Internal validation | High |
| Corral 2019 [40] | Risk Stratification | Path | 1 | 139 | 108 | 139 | ND | CV | MRI | CNN and SVM-based framework | AUC: 0.78 | Internal validation | High |
| Dmitriev 2021 [41] | Differential Diagnosis | Path | ND | 194 | 122 | 134 | 60 | TS | CT | CNN and RF-based framework | Acc: 91.7% | Internal validation | High |
| Gao 2020 [42] | Differential Diagnosis | Path | 2 | 504 | 53 | 398 | 106 | TS | MRI | GANs and transfer learning to train CNN; plurality voting | Acc: Patch Level: 71.56% (internal); 79.46% (external); Patient Level: 70% (internal); 76.79% (external) | External validation | Low |
| Hussein 2018 [43] | Risk Stratification | Path | ND | 139 | 108 | 139 | ND | CV | MRI | CNN and SVM-based framework | Acc: 64.67% (risk stratification); 82.80% (detection) | Internal validation | High |
| Hussein 2019 [44] | Detection | Rad | ND | 171 | 133 | 171 | ND | CV | MRI | Unsupervised: GIST, ∝SVM. Supervised: GIST, VGGfc7/8 features, RF & SVM | Acc: 58.04% (unsupervised - IPMN); 84.22% (supervised IPMN - VGGfc8+SVM) | Internal validation | High |
| Kuwahara 2019 [45] | Risk Stratification | Path | 1 | 50 | 50 | 50 | ND | CV | EUS | ResNet50 | AUC 0.98; Acc 94.0% | Internal validation | High |
| LaLonde 2019 [46] | Risk Stratification | Path | ND | 139 | 110 | 139 | ND | CV | MRI | Inflated neural networks; used modality fusion | Acc: InceptINN: 78.57% (early fusion); 75% (intermediate fusion); DenseINN: 75% (early fusion); 82.14% (intermediate fusion) | Internal validation | High |

*(Continued)*

**Table 1.** (Continued)

| Study ID | Prediction Targets | Ground Truth | Number of Centers | Number of Patients AI was Developed on | | Number of Patients | | Validation Procedure | Imaging Modality | Methodology | Performance Metrics | Stage of Clinical Translation | Overall ROB |
|---|---|---|---|---|---|---|---|---|---|---|---|---|---|
| | | | | Total | IPMN | Train | Test | | | | | | |
| Li 2019 [47] | Differential Diagnosis | Path | ND | 206 | 64 | 206 | ND | CV | CT | Dense-Net | Acc: 72.8% (overall); 81.25% (IPMN) | Internal validation | High |
| Liang 2022 [48] | Differential Diagnosis | Path | 1 | 94 | 39 | 94 | ND | CV | CT | RM & fused model using SVM and LR | AUC: 0.900 (RM); 0.973 (fused model) | Internal validation | High |
| Mazor 2023 [49] | Segmentation | Expert | 1 | 158 | 158 | 118 | 23 | TS | MRI | 3D Unet | Recall: 0.80 (detection); DSC: 0.80 | Internal validation | Unclear |
| Park 2023 [50] | Detection | Path | 2 | 2044 | 132 | 852 | 1192 | CV & TS | CT | nnUnet; ensemble voting | AUC: 0.91 (test set 1); 0.87 (test set 2) | External validation | Low |
| Qu 2023 [51] | Differential Diagnosis | Path | 1 | 221 | 8 | 147 | 74 | TS | CT | RM using deep neural network | AUC: 0.745; 0.624 (extended application) | Internal validation | High |
| Salanitri 2022 [52] | Risk Stratification | Path | 1 | 139 | 108 | 139 | ND | CV | MRI | Fine-tuned ViT; multimodal fusion | Acc: 0.70 (early fusion); 0.60 (late fusion) | Internal validation | High |
| Schulz 2023 [53] | Risk Stratification | Path | 2 | 70 | 70 | 43 | 27 | TS | EUS | Transfer learning to finetune CNN | Acc: 99.6% | External validation | Low |
| Shen 2020 [54] | Differential Diagnosis | Path | 1 | 164 | 48 | 115 | 49 | TS | CT | RM using SVM, RF, ANN | Acc: 71.43% (SVM); 79.59% (RF); 71.43% (ANN) | Internal validation | Unclear |
| Si 2021 [55] | Detection | Rad | 2 | 666 | 23 | 319 | 347 | TS | CT | CNN based architectures | Acc: 82.7% (overall); 100% (IPMN) | External validation | Unclear |
| Wang 2022 [56] | Risk Stratification | Path | 2 | 363 | 65 | 266 | 102 | CV & TS | CT | Traditional RM and a CNN | Acc: 0.750 (RM); 0.904 (DLM) | External validation | Low |
| Watson 2021 [57] | Risk Stratification | Path | 1 | 27 | 15 | 18 | 9 | CV & TS | CT | CNN based architecture | Acc: 8/9~88% | Internal validation | High |
| Yao 2023 [58] | Risk Stratification | Path & Rad | 5 | 246 | 176 | 197 | 49 | CV & TS | MRI | 3D nnUNet; ViT and RM | Acc: 81.9% | External validation | Unclear |
| Yuan 2023 [59] | Segmentation | Path & Expert | ND | 661 | ND | 378 | 118 | TS | CT | nnUNet with a transformer decoder | AUC 82.52% (outlier); DSC: 41.77% (inlier), 46.92% (IPMN) | Internal validation | Low |
| Zhang 2022 [60] | Differential Diagnosis & Risk Stratification | Path | 1 | 263 | 66 | 263 | ND | CV | CT | CNN and GNN | Acc: 88.92% (binary); 74.32% (4-classes) | Internal validation | High |

Ground Truth: Path: Pathological Diagnosis; Rad: Suspected by Radiologist; Expert: Segmentation by Radiologist. Validation Procedure: CV: Cross-Validation; TS: Test/ Validation Set. Methodology: RM: Radiomics Model; RF: Random Forest; SVM: Support Vector Machine; LR: Logistic Regression; GAN: Generative Adversarial Network; ViT: Vision Transformer; ANN: Artificial Neural Network; GNN: Graph Neural Network; DLM: Deep Learning Model. Other abbreviations: ND: Not Documented; ROB: Risk of Bias (PROBAST) Characteristic Curve; DSC: Dice Similarity Coefficient; DLM: Deep Learning Model. Other abbreviations: ND: Not Documented; ROB: Risk of Bias (PROBAST)

classifiers, achieving diagnostic accuracies between 67 and 94%. The remaining studies (n = 5) employed CNNs directly for image classification, with reported accuracies varying from 72 to 92%. For risk stratification, most studies implemented CNNs, with some incorporating image fusion techniques to integrate information from multiple MRI sequences. The average reported accuracy for these studies reached 85% (Fig 3C, Table 1, S2 Table, S1 Fig).

Risk of bias analysis using the PROBAST tool [34] identified methodological shortcomings in most included works on AI in IPMN imaging. More than half of the studies were identified as having a high risk of bias (n = 14, 56%), six with a low risk of bias (24%), and the remaining five with an unclear risk of bias (n = 5, 20%) (Table 1, S4 Table).

## Discussion

Recent advances in AI for analyzing medical imaging and imaging reports have demonstrated capabilities to improve clinical tasks such as diagnosis, prognosis, therapeutic planning, and monitoring disease progression [61]. Similarly, such advanced computational methods hold promise in improving the clinical management of IPMN by enhancing diagnostic accuracy and treatment selection. We conducted a systematic review of existing publications on AI in IPMN imaging, with an emphasis on evaluating their patient and data characteristics, methodology, application areas, and stages of clinical translation.

Our analysis indicates an increasing number of publications on AI-based IPMN diagnosis and risk stratification. Nonetheless, only a limited number of studies had a primary focus on IPMN [40,43–46,52,53,58]; instead, most works included smaller numbers of IPMN patients in addition to other patient cohorts, i.e., with other types of pancreatic cysts. A prevalent challenge identified across all the publications is limited data availability, as most studies trained the AI models on small, single-center datasets and did not publish data alongside the scientific analysis, resulting in an overall scarcity of high-quality PDAC datasets [62] and a notable absence of publicly available IPMN datasets. This is in contrast with the availability of datasets for other types of cancer, for instance, breast cancer [63,64] and brain tumor [65] imaging. International consortia and public-private partnerships are essential for addressing these limitations in data heterogeneity and the single-institution scope of studies, by systematic curation and sharing of high-quality datasets publicly. This will lead to better model training and rigorous validation across diverse patient populations, imaging parameters and clinical workflows. Moreover, the reporting quality appears inadequate as several publications do not document the exact number of centers involved, which compromises the validity and reproducibility of the study [66] as well as the adoption of new technology [67]. Our risk of bias analysis using the PROBAST tool [34] mirrored such methodological shortcomings in most included works on AI in IPMN imaging, with more than half identified as having a high risk of bias. Particularly frequent sources of bias were principally the lack of external validation, small cohort sizes, and unclear patient selection criteria, which may limit the generalizability of the findings. Adoption of standardized reporting frameworks, such as CLAIM for AI in medical imaging, can enhance methodological transparency and reproducibility of the studies [68].

The stratification of IPMN malignant progression based on imaging is challenging, with estimated diagnostic accuracies of imaging ranging from 60 to 80% in the scientific literature [69]. The established consensus guidelines for IPMN management and stratification recommend imaging, laboratory results, and physical examinations to stratify the risk of malignant progression [12,23]. However, the low specificity of existing methods [24] frequently leads to unnecessary surgery or surveillance of cystic lesions at high risk of malignancy. Computational analysis of pancreatic imaging can predict morphological types of IPMN noninvasively and bridge a crucial clinical gap by identifying patterns that could stratify and correlate with malignant progression in patients with IPMN. Most studies included in this systematic review proposing AI-based stratification of type of dysplasia based on imaging reported accuracies ranging from 60 to 99.6%, with most between 75 and 90% (Table 1, S2 Table).

We found CT to be the most frequently used imaging type. However, per the ACR's Appropriateness Criteria for pancreatic cysts, the preferred imaging modality for initial evaluation is an abdominal MRI without and with intravenous contrast with magnetic resonance cholangiopancreatography (MRCP). EUS is only recommended if worrisome features

are observed, and the cyst size is greater than 2.5 cm [35]. The preference for CT in literature is likely because it is more cost-effective, expeditious, and widely available in nearly all medical facilities [70], whereas MRI and EUS are protracted, costly, and comparatively less available [71]. It is noteworthy that the majority of studies rely solely on (one type of) imaging data; however, integrating multimodal clinical data could further improve model performance and enhance patient selection for surgical resection.

Methodologically, in deep learning-based studies, CNNs predominated in ROI segmentation and feature extraction, with UNet-based models, notably nnUNet [72], being the most frequently employed. In radiomics-based studies, which represented a relatively small proportion of studies investigating differential diagnosis, radiomic features (e.g., global histogram, first-order, second-order, wavelet features, etc.) are extracted and subsequently fed into classical machine learning algorithms (e.g., random forest and support vector machine) for prediction or classification. Withal, drawing definitive conclusions on the most effective AI methodology is challenging due to the dearth of large, high-quality data and heterogeneity in prediction targets. This observed heterogeneity reflects the absence of a large-scale benchmarking study to establish a preferred approach for AI-based prediction tasks in IPMN imaging. While few studies directly compare different approaches on the same task or dataset, most publications report the performance of a single model on a closed- or restricted-access dataset, complicating reproducibility, and comparison.

To address the recent developments of AI in IPMN imaging, our supplementary search identified seven additional studies published between February 2024 to May 2025, revealing several noteworthy trends. These studies utilized EUS [73,74], CT [75–78], and MRI imaging modalities [75,79], while maintaining the predominant focus on risk stratification tasks [73,75–77,79]. A particularly significant development is the emergence of federated learning approaches, demonstrated by Pan et al.'s multi-institutional study involving seven centers [79]. This represents an important step toward addressing the critical need for external validation while preserving data privacy. Additionally, we observed the application of advanced architectures such as Mamba YOLO v10 for detection tasks [78] and SegFormer [74] for segmentation applications. However, the persistent predominance of internal validation and single-center datasets highlights the lack of external validation that we identified in our primary analysis. Encouragingly, Cheng et al. progressed to prospective evaluation of their pancreatic cystic neoplasm risk stratification model [76], representing tangible movement along the translational pathway.

Models for language processing, particularly LLMs like GPT-4 [80] and Med-PaLM 2 [81], are increasingly being used in the field of medicine, owing to their potential to enhance patient care [82]. Despite our search strategy being inclusive of NLP and imaging reports, no such study met the inclusion criteria. However, the few retrieved studies employing language-based methods centered on the identification of patients with pancreatic cysts using rule-based algorithms [83–85]. One study used the pre-trained BioBERT model for extracting information from radiology reports in the context of pancreatic cysts, albeit without fine-tuning [86]. Therefore, the potential of modern LLMs in IPMN care, including vision-language models (VLMs), remains to be elucidated. VLMs, which combine imaging and textual data, could support clinical workflows such as automated radiology report generation and multimodal risk prediction models by aligning various imaging modalities with clinical notes. This can aid in standardizing the interpretation of imaging features of IPMN and linking them with guideline-based management. Optimization methods like Retrieval-Augmented Generation, could allow integrating authoritative knowledge such as clinical treatment guidelines in addition to its training data sources before generating outputs, may yield additional benefits over generalist LLMs in medical scenarios [87,88]. Moreover, agentic AI systems that can autonomously execute multi-step clinical workflows, such as continuously monitoring patient data, triggering alerts for guideline-based follow-up intervals, and coordinating care recommendations. This offers a path toward proactive and systematic oversight in IPMN management, by having AI agents as teammates to clinicians [89]. Future work should evaluate the feasibility of deploying such systems in real-world settings for management of IPMNs, where labeled datasets remain limited.

Despite various available guidelines, a standard PDAC screening examination for the general population has yet to be established. For example, routine screening mammograms are the most sensitive way to screen for breast cancer,

offering relatively affordable and accessible options for screening. Multiple AI-based breast cancer screening tools have received Food and Drug Administration (FDA) approval [90]. According to our review, only one study initiated a prospective clinical evaluation [38], and one study [41] extended the application to develop an end-to-end system [91] for the diagnosis of pancreatic lesions, which highlights the paucity of clinical translation and the absence of FDA-approved AI tools for screening IPMN and other pancreatic cysts. However, recent progress has emerged with PANDA developed by Cao et al. [38] receiving FDA breakthrough device designation, marking the first AI tool for PDAC detection to achieve this regulatory milestone. The disparity in regulatory approval between AI tools for cancers like breast and lung, and pancreatic applications likely stems from the relative rarity of pancreatic cancers compared to breast and lung cancer [92], which limits available training datasets and real-world evaluation [93]. Prospective observational studies are essential to evaluate real-world model performance in order to facilitate regulatory approval for clinical implementation. Furthermore, the integration of AI-based tool into the surgical decision-making for IPMN warrants specific ethical consideration, i.e., fairness, explainability and rigorous validation. Particularly in early translational phases, clinical decisions should not rely solely on the model's predictions; instead, AI outputs must be integrated thoughtfully into clinical workflows, with final decisions guided by clinician expertise. Ensuring that AI augments rather than replaces clinical standards is essential for ethical implementation of AI-based tools for IPMN management.

The limitations of this systematic review are largely related to the reporting standards of the included publications. Based on the large variation in study designs, prediction targets (Table 1), and reported performance metrics, an adequate summarization of model performances across all included studies was not feasible. We therefore report a complete list of metrics (Table 1, S2 Table) and demographic summary (S3 Table). We also limited our assessment of the reporting quality of individual studies to a general risk of bias assessment using the PROBAST [34] tool and a detailed review of the most relevant quality criteria for AI-based studies, including details on the patient cohort, methodological aspects, and the validation procedure. This is because established tools for assessing reporting quality in systematic reviews, such as QUADAS [94], are specifically designed for diagnostic accuracy studies and are therefore not appropriate for evaluating prediction model studies included in this review. The heterogeneity in our included study designs further necessitated the use of PROBAST's flexible framework, which can accommodate various prediction modeling approaches while maintaining standardized bias assessment criteria. Therefore, we could not perform a formal assessment of the reporting quality of the included studies.

In conclusion, this systematic review provides a comprehensive overview of the academic literature on AI in IPMN imaging and highlights the potential of computational image analysis models for the complex management of IPMN. Most existing studies rely on small, single-center patient cohorts and imaging data only. To exploit the full potential of AI for improving diagnostic accuracy and patient outcomes, it will be necessary to incorporate multicentric and multimodal clinical data and undertake robust efforts to translate retrospective research into clinical practice.

## Supporting information

**S1 Table. Data extraction template.**
(PDF)

**S2 Table. Data extracted from the included publications on AI in IPMN imaging.**
(PDF)

**S3 Table. Demographic characteristics of the included studies.**
(PDF)

**S4 Table. Risk of bias assessment using PROBAST Tool.**
(PDF)

**S1 Fig.  Performance Metrics Across Prediction Targets.** For detection, differential diagnosis, and risk stratification accuracy (Acc), area under the receiver operating curve (AUC), sensitivity (Sen), and specificity (Spec) are presented. For segmentation, the DSC is presented. — indicates that the metric was not reported.
(PNG)

**S1 Data.  PRISMA Checklist.**
(PDF)

**S2 Data.  AMSTAR Checklist.**
(PDF)

## Author contributions

**Conceptualization:** Muhammad Ibtsaam Qadir, Jackson A. Baril, Michele T. Yip-Schneider, Fiona R. Kolbinger.

**Data curation:** Muhammad Ibtsaam Qadir, Jackson A. Baril, Fiona R. Kolbinger.

**Formal analysis:** Muhammad Ibtsaam Qadir, Jackson A. Baril, Fiona R. Kolbinger.

**Funding acquisition:** Michele T. Yip-Schneider, C. Max Schmidt, Fiona R. Kolbinger.

**Investigation:** Muhammad Ibtsaam Qadir, Jackson A. Baril, Michele T. Yip-Schneider, Duane Schonlau, Thi Thanh Thoa Tran, C. Max Schmidt, Fiona R. Kolbinger.

**Methodology:** Muhammad Ibtsaam Qadir, Jackson A. Baril, Michele T. Yip-Schneider, Fiona R. Kolbinger.

**Project administration:** Michele T. Yip-Schneider, Thi Thanh Thoa Tran, C. Max Schmidt, Fiona R. Kolbinger.

**Resources:** Michele T. Yip-Schneider, C. Max Schmidt, Fiona R. Kolbinger.

**Software:** Muhammad Ibtsaam Qadir, Jackson A. Baril, Fiona R. Kolbinger.

**Supervision:** Michele T. Yip-Schneider, C. Max Schmidt, Fiona R. Kolbinger.

**Validation:** Muhammad Ibtsaam Qadir, Jackson A. Baril, Fiona R. Kolbinger.

**Visualization:** Muhammad Ibtsaam Qadir, Fiona R. Kolbinger.

**Writing – original draft:** Muhammad Ibtsaam Qadir, Fiona R. Kolbinger.

**Writing – review & editing:** Muhammad Ibtsaam Qadir, Jackson A. Baril, Michele T. Yip-Schneider, Duane Schonlau, Thi Thanh Thoa Tran, C. Max Schmidt, Fiona R. Kolbinger.

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
