## [Decision Letter · Decision Letter 0]

PDIG-D-24-00557Artificial Intelligence in Pancreatic Intraductal Papillary Mucinous Neoplasm Imaging: A Systematic ReviewPLOS Digital Health Dear Dr. Kolbinger, Thank you for submitting your manuscript to PLOS Digital Health. After careful consideration, we feel that it has merit but does not fully meet PLOS Digital Health's publication criteria as it currently stands. Therefore, we invite you to submit a revised version of the manuscript that addresses the points raised during the review process. Please submit your revised manuscript within 60 days Jul 07 2025 11:59PM. If you will need more time than this to complete your revisions, please reply to this message or contact the journal office at digitalhealth@plos.org. Please include the following items when submitting your revised manuscript:* A rebuttal letter that responds to each point raised by the editor and reviewer(s). You should upload this letter as a separate file labeled 'Response to Reviewers '. This file does not need to include responses to any formatting updates and technical items listed in the 'Journal Requirements' section below.* A marked-up copy of your manuscript that highlights changes made to the original version. You should upload this as a separate file labeled 'Revised Manuscript with Track Changes '.* An unmarked version of your revised paper without tracked changes. You should upload this as a separate file labeled 'Manuscript '. If you would like to make changes to your financial disclosure, competing interests statement, or data availability statement, please make these updates within the submission form at the time of resubmission. Guidelines for resubmitting your figure files are available below the reviewer comments at the end of this letter. We look forward to receiving your revised manuscript. Kind regards, Dhiya Al-Jumeily OBE, PhDSection EditorPLOS Digital Health Leo Anthony CeliEditor-in-ChiefPLOS Digital Healthorcid.org/0000-0001-6712-6626 **Additional Editor Comments (if provided):****Reviewers' Comments:** Reviewer's Responses to Questions

**Comments to the Author**

Reviewer #1: The systemic review is written in an intelligible fashion and concisely provides the research rationale and overall aim of the study. As well highlighted by the research, there is great need to improve the diagnosis of pancreatic cancer, while also avoiding surgical overtreatment. The paper describes novel applications of AI in IPMN imaging.

In terms of the revisions for this paper, I recommend minor revisions. Firstly, in the introduction of the paper, diagnostic errors (late or misdiagnosis of a disease) have been described but the key term not mentioned. Therefore, I would include this term and its definition, along with key statistics reported within the literature. For example, the number of diagnostic errors related to pancreatic cancer seen annually.

With regards to the methods, the systematic review has demonstrated the use of systemic review protocols and tools. Moreover, the systemic review protocol has been submits and registered with PROSPERO. Although individual studies have not been quality checked, the authors have explained rational for not carrying out individual assessments within the discussion section of this review. However, more detail should be given as to why the established quality tools were not suitable for the types of studies included.

Please mention within the 'Study Characteristics' section the type of studies included, such as case-control or cohort, retrospective or prospective. The authors can also include study characteristics such as the female to male ratio of participants, participants' age/age range and the country where the study took place if mentioned within the literature.

In table 1, the authors have identified that several of the papers demonstrated a high risk of bias, please highlight that these papers were approached with caution or further explain the inclusion of such papers.

The authors have made apparent heterogeneity within the methodologies of included studies, please explain why this variation was seen?

Within the discussion section, the authors have suggested a lack of accuracy of studies with low participants numbers. It is important to consider the generalisability of these studies.

Although figures 1 - 3 are highly informative and aid in the understanding of the research, the quality/resolution of images must been improved prior to publication.

A potential error was noticed within S3 Table. The Study by Zhang 2022 received low bias ratings for participants, predictors, outcome and analysis but received an overall bias rating of high.

Overall, this paper was intriguing and highlighted a novel application for AI of IPMN for the purpose of pancreatic cancer diagnosis.

Reviewer #2: The systematic review provides a solid foundation for understanding the role of AI in IPMN imaging. It is comprehensive in its examination of imaging modalities, prediction targets, and publication trends.

Reviewer #3: The paper entitled “Artificial Intelligence in Pancreatic Intraductal Papillary Mucinous Neoplasm Imaging: A Systematic Review” provide a systematic review addresses a timely and relevant topic by evaluating the current landscape of AI applications in imaging for pancreatic intraductal papillary mucinous neoplasms (IPMNs).

The paper is well-structured, comprehensive, and generally well-written. The authors rigorously applied PRISMA and AMSTAR guidelines and thoughtfully included a risk of bias assessment (PROBAST), which strengthens the methodological validity of the review. The inclusion of multiple imaging modalities (CT, MRI, EUS) and stratification of AI use cases (detection, segmentation, differential diagnosis, risk stratification) allows for a nuanced understanding of the field. The paper effectively identifies key research gaps, including small, single-center studies and limited clinical translation, while emphasizing the need for multimodal data and multicentre collaboration to advance AI in this field.

Although, the paper is reasonably structured and the topic is interesting, few issues are identified. Here are some issues and suggestions, that needs considering, to improve the quality of the manuscript:

1. Although the manuscript focuses on AI applications in IPMN imaging, many included studies do not centre exclusively on IPMN. Please clarify and justify the inclusion of studies where IPMN is a subset within broader pancreatic cyst or lesion cohorts. Consider stratifying the results more distinctly to delineate findings specific to IPMN.

2. While the manuscript reports various performance metrics (AUC, accuracy, etc.), more commentary on statistical rigor (e.g., confidence intervals, statistical significance, inter-rater agreement during data extraction) would enhance transparency. Were metrics statistically compared across models or studies?

3. The discussion would benefit from deeper analysis of specific regulatory, technical, or economic barriers that may explain the lack of FDA-approved AI tools for IPMN, in contrast with other cancers such as breast or lung.

4. Although the paper states that data is available in the manuscript and supplementary files, it’s unclear if raw data or code used in the review (e.g., data extraction sheets, PROBAST assessments) are available for reproducibility. Consider depositing these in a public repository.

5. The manuscript is generally intelligible and written in standard academic English. However, a few grammatical issues and overly complex sentence constructions in the introduction and discussion sections could be revised for better readability. For example, the sentence beginning "Given the substantial risks associated with pancreatic surgery..." is quite dense and could be simplified.

6. Consider including a summary table or heatmap of PROBAST scores across studies in the main text to better visualize common sources of bias across domains (participants, predictors, outcomes, analysis).

7. While heterogeneity precludes a formal meta-analysis, the inclusion of a weighted average performance score (e.g., average AUC or accuracy across use cases) or a summary figure of model performance could help readers synthesize the findings quantitatively.

8. The discussion introduces potential applications of large language models (LLMs), such as GPT-4 and Med-PaLM 2, but lacks depth. A more critical exploration of how vision-language models could be integrated into IPMN workflows would add a forward-looking component.

9. While the review is focused on technical aspects, it may be worthwhile to briefly discuss ethical considerations in applying AI to high-stakes surgical decision-making, especially given the overtreatment risks in IPMN.

10. Some terms and abbreviations (e.g., RM, ViT, DLM) are introduced without clear definition in the text. Consider including a glossary or ensuring all are defined at first use.

Reviewer #4: The authors conducted a systematic review to assess the potential of AI-based methods in enhancing diagnosis and risk stratification for intraductal papillary mucinous neoplasm (IPMN). They categorized the selected studies based on prediction targets, imaging modality and data type, patient cohort size, and stage of clinical translation. A major limitation identified across studies was the scarcity of large, high-quality datasets, particularly multicenter and publicly available ones.

The paper is written well but needs some improvements before publication:

- Considering the rapid development of AI in medical imaging, it would be beneficial for the authors to include studies published in 2024. This would ensure that the review remains current and does not miss recently developed models or advancements that might significantly impact the field by the time of publication.

- While the paper clearly identifies major limitations such as the lack of high-quality, large-scale datasets and limited clinical translation, it would strengthen the manuscript if the authors could offer concrete suggestions or discuss potential strategies to overcome these challenges.

Reviewer #5: General Comments

This systematic review provides a timely and comprehensive synthesis of the current state of AI applications in IPMN imaging. The manuscript is well-structured, adhering to PRISMA and AMSTAR guidelines, and addresses a clinically relevant gap in the specificity of IPMN management. The authors effectively highlight the potential of AI to mitigate overtreatment and improve diagnostic accuracy. However, several methodological and translational limitations warrant discussion to strengthen the manuscript.

Recommendations for Authors

1. Strengths

• Scope and Rigor

The review encompasses 25 studies (2018–2023), offering a robust analysis of AI methodologies (e.g., CNNs, radiomics) and imaging modalities (CT, MRI, EUS). The stratification by prediction targets (detection, segmentation, diagnosis, risk stratification) is particularly insightful.

• Clinical Relevance

The discussion on the limitations of Fukuoka/Kyoto guidelines and the role of AI in enhancing specificity is compelling. The emphasis on multicenter collaboration and multimodal data integration aligns with unmet clinical needs.

• Transparency

Prospective registration (OSF, PROSPERO) and detailed risk-of-bias assessment (PROBAST) enhance reproducibility.

2. Limitations and Suggestions for Improvement

• Heterogeneity of Studies

- The wide variability in cohort sizes (e.g., 27 to 9,939 patients), imaging modalities, and performance metrics (e.g., AUC: 0.624–0.996) complicates cross-study comparisons. A meta-analysis might be unfeasible, but a qualitative synthesis of high-performing models (e.g., nnUNet for segmentation) could be expanded.

- Suggestion: Include a table summarizing top-performing models per use case with key metrics (e.g., sensitivity, specificity) and dataset characteristics.

• Data Availability and Bias

- Most studies used single-centre, retrospective datasets with unclear selection criteria, raising concerns about generalizability. Only 32% of studies underwent external validation.

- Suggestion to discuss potential biases (e.g., selection, spectrum bias) and advocate for standardized reporting (e.g., STARD-AI, TRIPOD-AI) in future studies.

• Underrepresentation of EUS and Multimodal AI

- Only 8% of studies utilized EUS, despite its clinical utility for worrisome features. The lack of NLP/LLM applications is notable.

- Suggestion: Address this gap in the discussion, proposing EUS-based AI and multimodal (imaging + clinical data) approaches as future directions.

• Clinical Translation

- Only one study (Cao et al., 2023) reached prospective clinical evaluation. The absence of FDA-approved tools contrasts with AI advancements in other cancers (e.g., breast).

- Suggestion: Expand the discussion on barriers to translation (e.g., regulatory hurdles, clinician trust) and potential solutions (e.g., real-world validation frameworks).

3. Minor Edits

• Clarify Metrics: In Figure 3C, specify if reported metrics are median/mean values and their ranges.

• Funding Statement: The NIH grant is appropriately disclosed but clarify if funders influenced study design/outcomes.

• References: Ensure all citations (e.g., PROBAST, QUADAS) are consistently formatted.

---

## [Editor Report · Decision Letter 1]

Artificial Intelligence in Pancreatic Intraductal Papillary Mucinous Neoplasm Imaging: A Systematic Review

PDIG-D-24-00557R1

Dear Dr. Kolbinger,

We are pleased to inform you that your manuscript 'Artificial Intelligence in Pancreatic Intraductal Papillary Mucinous Neoplasm Imaging: A Systematic Review' has been provisionally accepted for publication in PLOS Digital Health.

Best regards,

Dhiya Al-Jumeily OBE, PhD

Section Editor

PLOS Digital Health

**Additional Editor Comments (if provided):**

Comments have been addressed to satisfactory level in the revised version.
